# Echolocating bats prefer a high risk-high gain foraging strategy to increase prey profitability

Laura Stidsholt[1]*, Antoniya Hubancheva[2,3], Stefan Greif[2,4], Holger R Goerlitz[2], Mark Johnson[1], Yossi Yovel[4], Peter T Madsen[1]

[1]Zoophysiology, Department of Bioscience, Aarhus University, Aarhus, Denmark; [2]Acoustic and Functional Ecology, Max Planck Institute for Biological Intelligence, Seewiesen, Germany; [3]Department of Animal Diversity and Resources, Institute of Biodiversity and Ecosystem Research, Bulgarian Academy of Sciences, Sofia, Bulgaria; [4]Department of Zoology, Tel Aviv University, Tel Aviv, Israel

**Abstract** Predators that target multiple prey types are predicted to switch foraging modes according to prey profitability to increase energy returns in dynamic environments. Here, we use bat-borne tags and DNA metabarcoding of feces to test the hypothesis that greater mouse-eared bats make immediate foraging decisions based on prey profitability and changes in the environment. We show that these bats use two foraging strategies with similar average nightly captures of 25 small, aerial insects and 29 large, ground-dwelling insects per bat, but with much higher capture success in the air (76%) vs ground (30%). However, owing to the 3–20 times larger ground prey, 85% of the nightly food acquisition comes from ground prey despite the 2.5 times higher failure rates. We find that most bats use the same foraging strategy on a given night suggesting that bats adapt their hunting behavior to weather and ground conditions. We conclude that these bats use high risk-high gain gleaning of ground prey as a primary foraging tactic, but switch to aerial hunting when environmental changes reduce the profitability of ground prey, showing that prey switching matched to environmental dynamics plays a key role in covering the energy intake even in specialized predators.

*For correspondence: laura.stidsholt@bio.au.dk

Competing interest: The authors declare that no competing interests exist.

## Editor's evaluation

This study presents important findings on the hunting strategies and energy intake of bats in the wild. It combines several methods (biologging, captive experiment, and DNA metabarcoding) to provide convincing evidence for the claims. With detailed data and analyses on foraging ecology, this work will be of broad interest to animal ecologists.

## Introduction

For many predators, the ability to switch between multiple prey types is key to surviving dynamics in prey availability. While some prey types are only available sequentially e.g., over seasons, others are available simultaneously and predators must choose when to switch between them. In these situations, predators are predicted to ignore low profitability prey when more profitable prey are present and abundant, and only switch prey type if there is a perceived prospect of increased profitability (i.e. more energy gained per time unit of hunting) (*Stephens and Krebs, 1986*). However, testing such fundamental predictions of prey switching in the wild are greatly complicated by the difficulties of measuring encounter rates and capturing successes to estimate prey profitability of individual predator-prey interaction (*Sih and Christensen, 2001*). Here, we use a biologging approach on

**eLife digest** Bats are the only mammals capable of powered flight and therefore need a high calorie intake to survive. They hunt at night using the echoes made by their own calls to navigate and locate prey.

Bats can use different tactics to hunt for food: hawking involves catching prey on the wing and requires fast aerial manoeuvring and more intense echolocation calls, while gleaning involves listening for movements of ground and water dwelling prey as the bat hovers. Some bat species specialise as hawkers or gleaners but maintain the ability to hunt with both methods. With the ever-growing impact of human activities on their habitats, it is important to understand how adaptable bats feeding habits are to changes in their environment.

To find out more, Stidsholt et al. studied greater mouse-eared bats, which primarily feed by gleaning. To understand how this species chooses feeding strategies they fitted bats with tiny back-packs that could record the animal's location and foraging behaviour. They could also monitor prey sizes by recording the sounds of the bats chewing.

Stidsholt et al. found that, although these bats tried to catch prey on the ground more often than in the air, they were actually more successful as airborne hunters. Despite this, gleaning was still a more profitable strategy for them, because the body mass of ground prey is higher than for airborne prey. Gleaning gave the bats a higher calorie intake, even though their capture rate was lower.

Although feeding habits differed slightly between individual bats on a given night of monitoring, there were much larger changes in behaviour between different feeding nights. This shows that, although this species of bat prefers gleaning, they will switch strategies to hawking as their environment changes, for example if there is more airborne prey, or if rainfall makes it hard to hear movements on the ground. Bats tended to get enough calories for their needs but did not catch more prey than they needed to survive.

Stidsholt et al. concluded that greater mouse-eared bats change their feeding strategy based on prey availability and size, as well as the bat's environment. Their study provides an important insight into how bats fit into the ecosystem and how adaptable bats might be to changes in their habitat.

echolocating bats as a model organism to investigate how prey profitability influences the foraging strategies of a wild predator known to target prey inhabiting two different habitats (*Arlettaz, 1996*).

Bats are widespread and abundant predators that serve important roles in ecosystems across the globe (*Kunz et al., 2011*). Their evolutionary success is due to the unique combination of echolocation and powered flight (*Teeling et al., 2005*) allowing them to avoid visual predators by feeding at night, thereby gaining unfettered access to food sources that include insects, small vertebrates, fruit, nectar, pollen, and blood (*Simmons, 2005*). Within the mosaic of foraging niches they exploit, echolocating bats are categorized into three main foraging strategies: the ancestral mode of capturing prey on the wing (hawking), and the derived modes of trawling prey from water surfaces or gleaning prey, nectar, or fruit from ground and trees (*Schnitzler and Kalko, 2001*). To aid these specialized hunting strategies, each guild of bats has evolved specific adaptations in echolocation signals, auditory systems, morphology, and flight mechanics (*Norberg and Rayner, 1987*; *Fenton, 1990*; *Schnitzler and Kalko, 2001*). Despite such specialism, recent research has shown that foraging style is not monotypic within species: gleaning bats occasionally capture aerial prey (*Bell, 1982*; *Fenton, 1990*; *Ratcliffe and Dawson, 2003*; *Ratcliffe et al., 2006*; *Hackett et al., 2014*), while insect-gleaning bats may seasonally target nectar or fruit (*Aliperti et al., 2017*) or vice versa (*Herrera M. et al., 2001*). These changes in foraging style presumably track the relative abundance of preferred versus alternative food sources, broadening the ecological roles of bats and providing a degree of resilience in the face of changing resources. However, owing to the complexity of studying detailed hunting behaviors in the wild, it is not clear why or when specialized bats switch foraging strategies.

To address this, we asked whether bats adapt their hunting strategies continuously to maintain net intake or if switching is the last resort when preferred prey are unavailable. To do so, we used miniaturized biologging devices to track the hunting behavior of greater mouse-eared bats. This species primarily captures ground-dwelling arthropods by passively listening for their movements (i.e. gleaning, *Video 1*; *Video 2*; *Arlettaz, 1996*), and is, therefore, specialized for gleaning: broad, short

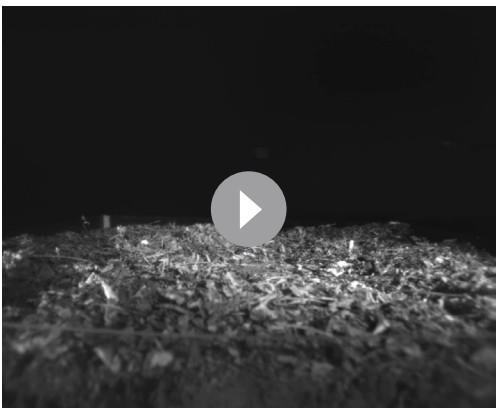

**Video 1.** A bat captures prey on the ground in a forest floor reconstructed in a flight room while carrying a tag. We used these laboratory experiments to ground truth in the wild data.

https://elifesciences.org/articles/84190/figures#video1

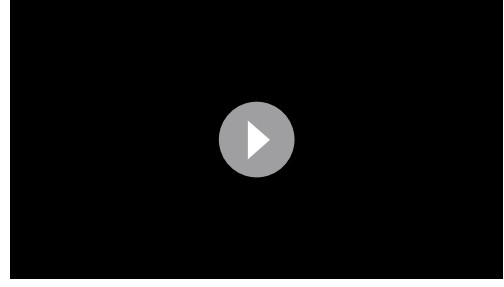

**Video 2.** Synchronized audio and acceleration data of a ground capture in the wild. Here, the mating song of the targeted bush cricket is audible and can be clearly seen in the spectrogram.

https://elifesciences.org/articles/84190/figures#video2

wings enable their take-off from the ground, and weak echolocation calls avoid alerting prey while also allowing the bat to hear rustling sounds of surface-dwelling prey. Nonetheless, like many other gleaning bats, greater mouse-eared bats have maintained the ancestral ability to use echo-location to capture aerial prey on the wing (i.e. hawking, *Video 3*; *Video 4*; *Video 5*; *Stidsholt et al., 2021*), requiring that they switch to intense calls to detect small prey, and maintain the capability to maneuver fast in 3D space to track evasive prey. While call intensity can be adjusted to fit different strategies, the morphological and anatomical specializations for ground gleaning must affect the efficiency of these bats as aerial hawkers. We, therefore, predicted that these bats would prefer gleaning whenever it was profitable, and would only switch to aerial hunting when environmental conditions led to poor energy intake rates when gleaning. Specifically, we tested the hypotheses that (1) although sensorimotor adaptation to gleaning comes at the cost of a reduced ability to capture aerial prey, bats continue to rely on both foraging strategies to cover their energy intake; (2) bats prefer the foraging strategy with the highest prey profitability; and (3) bats maintain their energy intake by adapting their foraging strategies to the habitat and environment. To test these hypotheses, we used miniaturized biologging devices to track the hunting behavior of 34 greater mouse-eared bats (*Stidsholt et al., 2019*). These tags recorded the bats' echolocation behavior, three-dimensional movement patterns, GPS locations (N=7 of 34 bats) and mastication sounds after prey captures *Video 6* as a proxy for foraging success. We complemented these data with DNA metabarcoding of feces from co-dwelling con-specifics (N=54 bats) to identify prey species and sizes. This combined biologging and DNA metabarcoding approach allowed us to quantify strategy

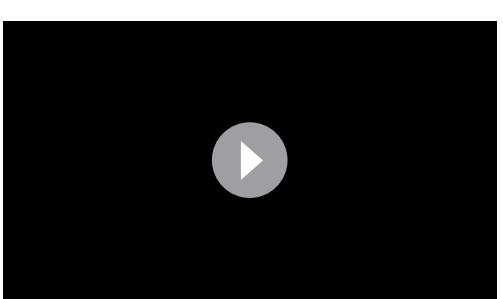

**Video 3.** A trained greater mouse-eared bat captures a tethered moth. We used these video recordings to ground truth the data from the wild.

https://elifesciences.org/articles/84190/figures#video3



**Video 4.** One minute of aerial hunting by a wild greater mouse-eared bat. The 3D flight pattern is reconstructed from the sensor data on the tag and is shown in white. Each aerial prey attack is marked as circles color-coded according to foraging success (green = success vs red = failure).

https://elifesciences.org/articles/84190/figures#video4



**Video 5.** Aerial capture in the lab with a tag. We trained bats to catch tethered moths or mealworms with and without tags in a dark flight room while filming their behavior with an infrared camera.
https://elifesciences.org/articles/84190/figures#video5

switching in a wild predator as a function of prey profitability and habitat.

## Results

To categorize the foraging strategies used by wild bats, we analyzed one night of sound and movement data from each of 34 female, greater mouse-eared bats (*Myotis myotis*). All bats commuted from the colony or release site to one or several foraging grounds before returning back to the roost before dawn (*Figure 1ABC*). Since, we used two different types of biologging devices with different sensors, foraging bouts were defined either as flight intervals of more than 50 s with a high variation in heading based on accelerometer and magnetometer data, or as intervals of an area-restricted search for tags including GPS (N=7 bats).

### Foraging success and number of prey attacks according to foraging strategy

A total of 3917 attacks on prey (*Figure 1D,E*) were recorded with most bats capturing prey both on the ground by passively gleaning prey (*Figure 1—figure supplement 1*), and by pursuing prey mid-air by aerial hawking (*Figure 1—figure supplement 2*). However, four bats exclusively gleaned, while two bats only hawked (*Figure 1D,E*). The dominant foraging strategy used per bat per night seemed to be affected by the night of tagging indicating that bats tagged on the same nights choose the same foraging strategy (*Figure 1D,E*, *Figure 1—figure supplement 3*, N=10 nights, 1–9 bats tagged per night; LMM; testing if the ratio between ground:aerial captures was explained by the night of tagging *Supplementary file 1g*).

The bats attacked food more often on the ground (mean: 80 attacks per individual per night, quartiles: 26–110) than in the air (mean: 36 attacks, quartiles: 7–70; GLMM, Poisson distribution, p=0.002, *Supplementary file 1*, *Figure 1D*), but the proportion of attacks that were successful (i.e. success ratio) based on audible mastication sounds following prey captures were more than double in the air (mean: 76%, quartiles: 71–88) than on ground (30%, quartiles: 25–40; *Supplementary file 1*, *Figure 1F*, *Figure 1—figure supplement 4*). This led to on average 25 (quartiles: 10–33) aerial and 29 (quartiles: 5–53) ground insects caught per bat per night (*Figure 1D*). The bats attacked prey more often on the ground (35 s between captures, quartiles: 17–70 s) compared to in the air (51 s between attacks, quartiles: 31–111, LMM, p<0.00001) (*Figure 1G*). Thus, bats caught prey much more reliably in the air, but attacked ground prey more often and devoted more foraging time to ground gleaning.

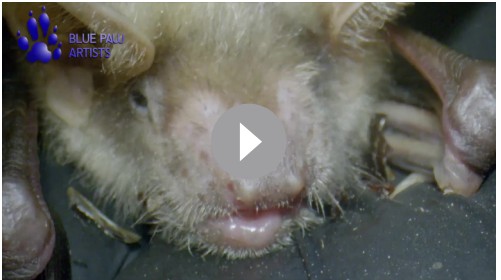

**Video 6.** Mastication of a bat is used to determine success ratios and prey sizes. The audible mastication sounds of the greater mouse-eared bats were used to determine successful prey attacks, and to measure the relative prey sizes between foraging strategies.
https://elifesciences.org/articles/84190/figures#video6

### The effect of habitat on foraging strategies

We next used GPS tracks from seven bats to investigate the behavioral and ecological factors that influence foraging success. Specifically, we tested if movement style (i.e. commuting vs actively searching for prey defined via the Lavielle method *Hurme et al., 2019*, *Figure 2—figure supplement 1*) and habitat (i.e. forest vs open fields, *Figure 2*) affected the success ratio and prey attacks. The GPS-tagged bats also predominantly caught prey in separate foraging bouts that were each dedicated to either hawking (*Figure 2A*, blue diamonds) or gleaning (*Figure 2A*, green circles) (*Figure 2—figure supplement 2*).

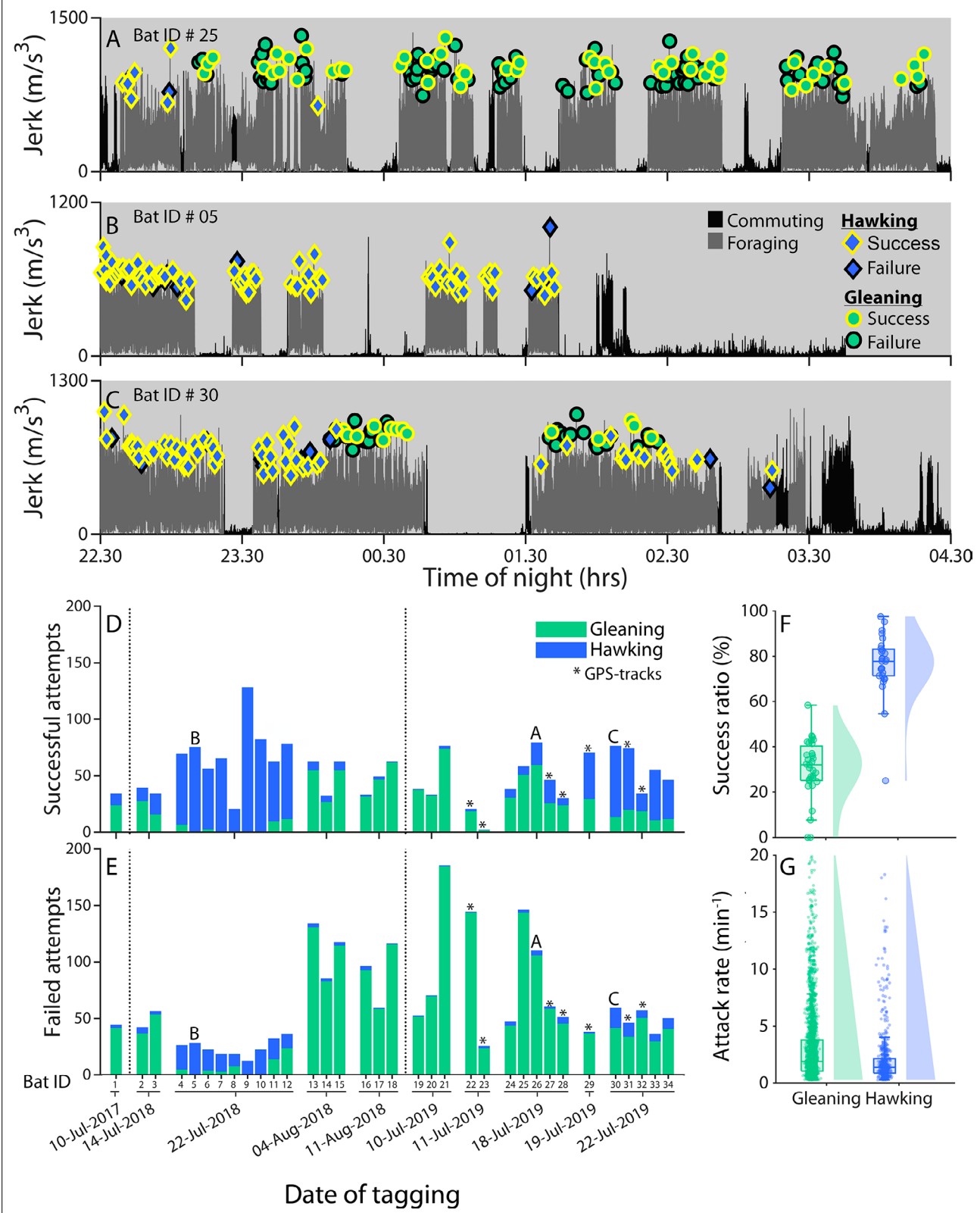

**Figure 1.** Greater mouse-eared bats tagged on different nights show wide variation in foraging strategy and success. (**A-C**) The jerk (differential of on-animal recorded acceleration) reveals the overall movement of the bat by showing periods of no movement (rest) and strong movement (flight) for three different bats (summed values for Bat ID 25, 5, and 30 as depicted in panel **D**) and two different travel modes (commuting (dark gray) vs foraging (light gray)). We marked all prey attacks as either hawking (blue diamonds) or gleaning (green circles) by visual and auditory inspection of the sound and

*Figure 1 continued on next page*

*Figure 1 continued*

movement data. Prey captures were classified by audible mastication sounds as successful (yellow edge) or failures (black edge). The bats exemplified here either primarily gleaned (**A**), primarily hawked (**B**), or used both strategies in alternating bouts (**C**). ( **D–E**) Successful (**D**) and unsuccessful (**E**) prey attacks of all bats (N=34) grouped according to the night of tagging for aerial hawking (blue) and gleaning (green). Stars mark the bats equipped with GPS tags; A, B, and C mark the bats depicted in panels **A-C**. (**F–G**) The success ratio (**F**) reveals the percentage of all attacks that were successful per bat per night (dots), while attack rates (**G**) reveal the number of foraging attacks per minute for each bat per night (dots) with more than one prey attack per foraging strategy for aerial hawking (blue) and gleaning (green) along with kernel densities and boxplots.

The online version of this article includes the following figure supplement(s) for figure 1:

**Figure supplement 1.** Verification of ground capture attacks in the wild.

**Figure supplement 2.** Differences between automatic and manual detection of aerial prey capture.

**Figure supplement 3.** The night of tagging affected the dominant foraging strategy of each bat.

**Figure supplement 4.** Verification of the automatic detector used to detect mastication sounds after prey capture attempts in the wild.

However, almost half of all aerial prey was captured during commuting (47% of total aerial captures; *Figure 2—figure supplement 1*). When gleaning, the bats attacked the same total number of prey in forest and open field habitats (field: 207 vs forest: 221 attacks in total, *Figure 2C*), but with more attacks per bout when gleaning above fields (field: 25 vs forest: 14 attacks/bout). Moreover, gleaning in open fields was twice as successful as in forest habitats (success ratio of 48% per foraging bout with more than two prey attacks in open fields vs 12% in the forest, *Figure 2E*, *Supplementary file 1*). In contrast, when capturing insects in the air, the attack rates and success ratios were consistently high and unaffected by habitat (*Figure 2DF*, LMM, *Supplementary file 1*).

## Estimation of prey sizes

To estimate prey types and sizes, we first performed DNA metabarcoding analysis on the feces of 54 untagged greater mouse-eared bats from the same colony caught in the morning upon returning from the foraging grounds. The bats target a wide range of prey species spanning 155 OUT (Operational Taxonomic Units) (*Figure 3AB*), of which ~60% occupy aerial niches (36 families), and ~40% occupy ground niches (23 families; *Figure 3A*). Ground prey was 2.5 x longer than aerial prey (20 mm (quartiles: 14–28 mm) vs 7 mm (quartiles 4–9 mm), *Figure 3C*) estimated from measured lengths of representatives of each species from online photo databases covering the same region in Bulgaria. We used length-weight regressions (*Straus and Avilés, 2018*) for each group to convert body-length to body mass for each prey type. For this analysis, we used the weighted average of the two most numerous prey types for aerial and ground regression values. Under these assumptions, estimated dry body masses of ground prey were ~20 times heavier than aerial prey (means: 67.5 mg quartiles: 29.7–146.6 mg) vs 3.0 mg (quartiles: 0.7–5.7 mg, *Figure 3E*, green and blue circles).

Since DNA metabarcoding does not provide the exact proportion of caught prey items and species, and thus does not allow to calculate the size distribution of caught prey, we performed an analysis of the mastication sounds as an additional proxy for prey size for nine bats with the best signal to noise ratio of the audio data (Tag type A). Greater mouse-eared bats chew all prey while flying irrespective of how they are caught and take longer to masticate larger prey (verified in laboratory feeding experiments, *Figure 3—figure supplement 1*). We used body length-to-body mass conversions from the DNA metabarcoding of ground prey as the reference prey body mass. We then estimated aerial prey body masses from the difference in chewing durations between ground and aerial prey. Bats chew longer on ground prey than aerial prey, indicated by the ~3 x more mastication sounds detected after each gleaning capture (75, quartiles: 38.5–111.5, *Figure 3D*) compared to aerial hawking (23, quartiles: 15–55). By applying this ratio to the body mass estimations of gleaning prey, we estimated aerial prey body mass of 21.2 mg on average (quartiles: 9.3–45.9) (*Figure 3E*, blue triangles). Thus, in the following, we use both a lower and a higher estimate of aerial prey body masses of 3.0 mg (from DNA metabarcoding), and 21.2 mg (from masticating sounds) (*Figure 3E*). Taking successful prey captures into account and the weighted average of the caloric values of the two most numerous prey types for aerial and ground (25.4 kJ/g dry mass of ground prey *Bell, 1990*; *Zygmunt et al., 2006*) and 21.3 kJ/g dry mass of aerial prey (*Kurta and Kunz, 1987*; *Bell, 1990*), the bats ingested an average of 60.2 kJ/night/bat (quartiles: 32.1–84.8) based on the lower aerial prey body mass estimates (*Figure 3G*, solid gray line), and 74.9 kJ/night/bat (quartiles: 55.2–95.7) based on the higher estimates (*Figure 3G*,

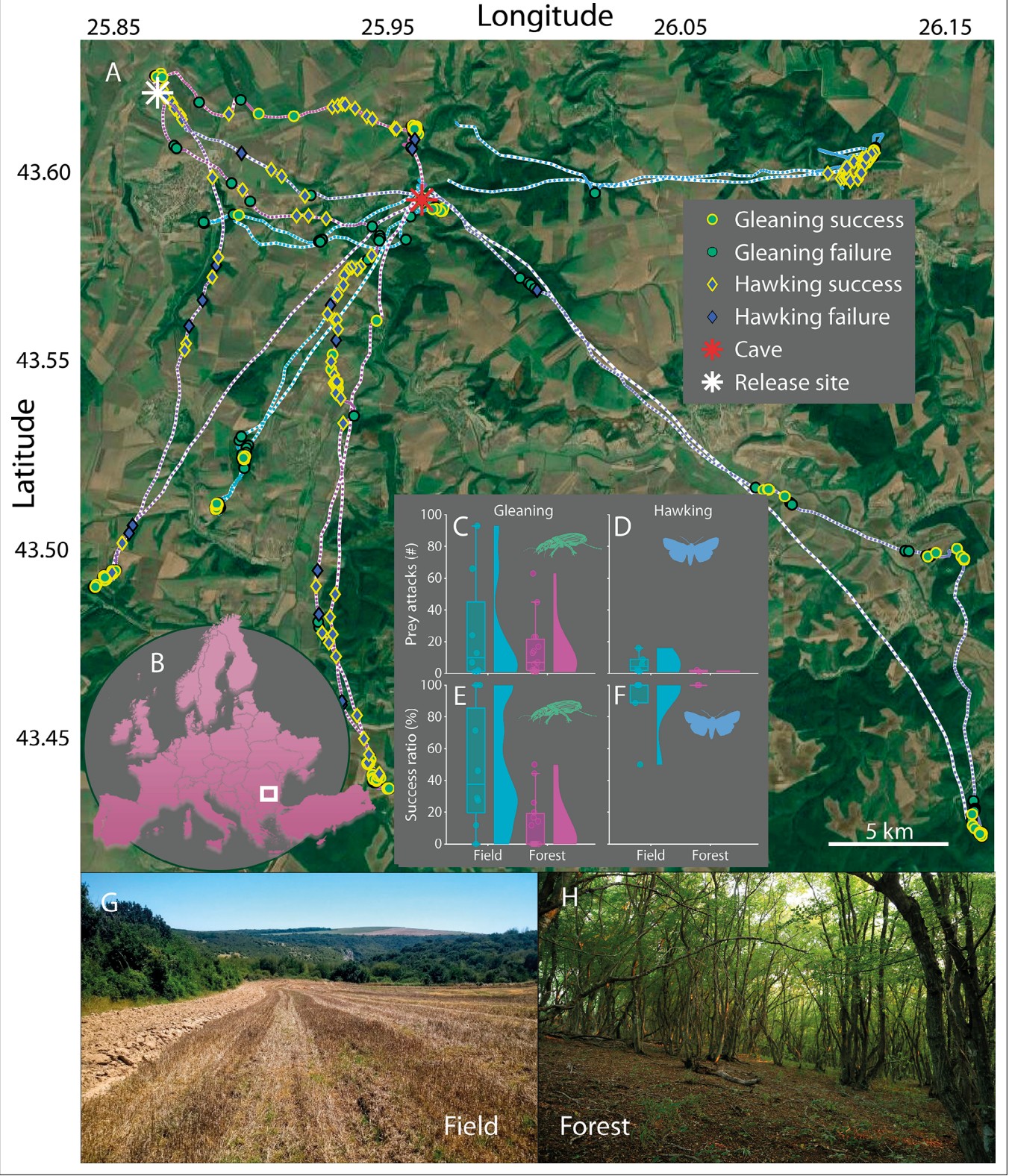

**Figure 2.** Habitat influences the foraging success of greater mouse-eared only when gleaning. (**A**) Tracks of seven bats with GPS tags released either at the cave (red star) or at a location nearby (white star) and their foraging behavior: Gleaning (green circles) and hawking (blue diamonds) attacks along with their success (yellow edge) or failure (black edge). (**B**) The bats were tracked in North-Eastern Bulgaria (white square). C-F: Total prey attacks (CD) and success ratios per foraging bout (EF), for both habitats: open field (blue; **G**) and forest (magenta; **H**). Each data point corresponds to one foraging bout. G-H: The two main foraging habitats of greater mouse-eared bats: open fields (**G**) and the open spaces below the canopy in forests (**H**).

*Figure 2 continued on next page*

*Figure 2 continued*

The online version of this article includes the following figure supplement(s) for figure 2:

**Figure supplement 1.** Foraging success and prey attack rates according to movement style.

**Figure supplement 2.** The bats mainly forage in dedicated foraging bouts.

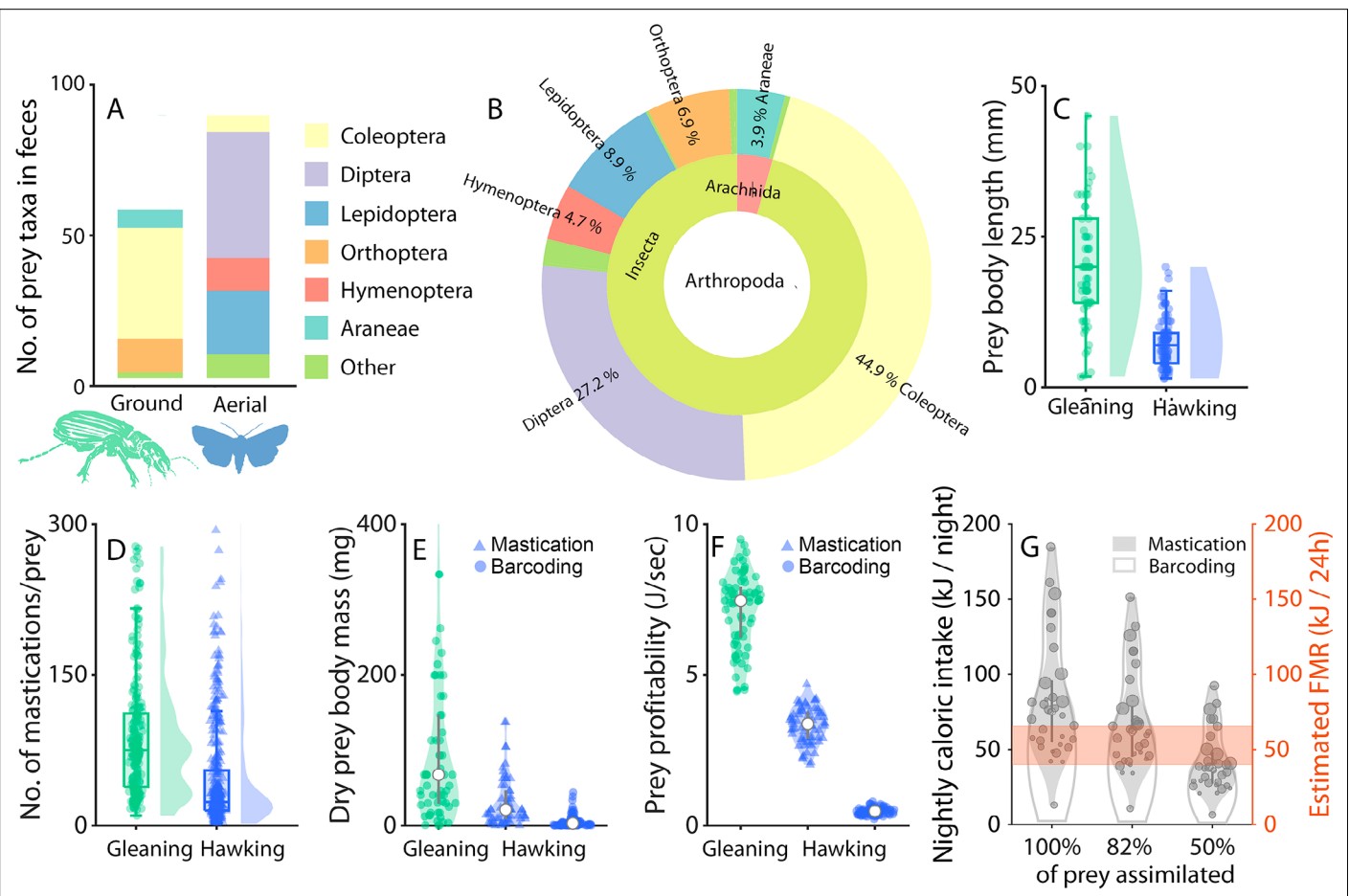

**Figure 3.** Ground prey is larger than aerial prey and sufficient to offset the lower foraging success ratios of gleaning. (**A-B**) DNA metabarcoding of feces from 54 greater mouse-eared bats (48 females, six males). Insects were categorized as either ground (green) or aerial (blue). The few prey species (N=5) that are both aerial and ground were omitted from the analysis. Distribution of the targeted prey orders depicted as OTU (Operational Taxonomic Units) between ground (~40%) and aerial (~60%) niches (**A**) and across taxonomical units in the Arthropoda (**B**). (**C-F**): Prey properties and profitability during gleaning (ground prey, green) and aerial hawking (aerial prey, blue), with kernel densities and boxplots. (**C**) Body lengths of the prey sorted by foraging strategy. (**D**) Number of mastication sounds identified after each prey capture by an automatic detector (N=244 ground captures and 336 aerial captures across 10 bats). (**E**) Dry prey body masses of each prey type identified for gleaning via DNA metabarcoding (green circles) (DNA metabarcoding was used as the reference prey body mass for ground captures), and for aerial prey by mastication analysis (blue triangles) and DNA metabarcoding (blue circles). (**F**) Prey profitability of gleaning or hawking prey calculated from prey body masses from mastication analysis (triangles) or DNA metabarcoding (circles) combined with observed success ratios, handling, and search times (*Figure 3—figure supplement 2*). The data are plotted for bootstrapped data (N=70 random data points) due to varying sample sizes of each parameter. (**G**) Total caloric intake per night per bat calculated by multiplying the caloric intake per prey with the number of successful gleaning and hawking prey captures (*Figure 1*), and compared to the field metabolic rate of a 30 g bat estimated from the literature (*O'Mara et al., 2017*) (orange).

The online version of this article includes the following figure supplement(s) for figure 3:

**Figure supplement 1.** Mastication duration change with prey size.

**Figure supplement 2.** The handling time of the prey caught either by gleaning or hawking is similar.

**Figure supplement 3.** Wingbeat rate and strength are similar across foraging strategies.

shaded gray). Using energy assimilation rates of 50–82% in bats (*Kurta et al., 1989*; *Straus and Avilés, 2018*), the bats obtained on average between 30–61 kJ/night per bat (*Figure 3G*) during the hunting seasons of July to August.

### Prey profitability

The profitability of prey caught by gleaning or hawking for all bats (N=34) was quantified by combining success ratios (*Figure 1F*), and search and handling times (*Figure 3—figure supplement 2*) with lower (*Figure 3E*, circles) and higher estimates of prey body masses (*Figure 3E*, triangles). The gleaning foraging strategy yielded a profitability of 7.4 J/s (quartiles: 5.6–8.1 J/s, *Figure 3F* green), while hawking resulted in a lower estimate of 0.5 J/s (quartiles: 0.4–0.54, *Figure 3F*, blue circles) and a higher estimate of 3.3 J/s (quartiles: 2.8–3.8, *Figure 3F*, blue triangles). Prey profitability when gleaning is thus 2.3–14 times higher than when aerial hawking.

## Discussion

The small size and high metabolic rate of bats out of hibernation, coupled with a costly locomotion mode, require a high and constant input of energy from foraging (*Kleiber, 1947*). This, in turn, calls for either stable and narrow food niches or adaptive hunting behaviors that track habitat and prey dynamics. Here, we used biologging and metabarcoding to explore how greater mouse-eared bats chose between two different foraging strategies to cover their high energy intake, and how strategy switching is adapted to habitat and environment.

### Greater mouse-eared bats are more successful when hawking, but gain more energy from gleaning prey off the ground

Since greater mouse-eared bats are a gleaning specialists, we first hypothesized that their sensorimotor adaptation to gleaning would come at the cost of a poorer ability to capture aerial prey during hawking, and that they, therefore, would rely mostly on the ground foraging to cover their energy intake. Our finding that the tagged bats were less successful in gleaning prey from the ground compared to hawking insects mid-air (30 vs 76% success ratio) (*Figure 1F*) leads us to dismiss that hypothesis. Surprisingly, the high success ratios for greater mouse-eared bats when hawking are on par with observations in the wild for hawking specialist bats (*Rydell et al., 2002*). Such high success rates are likely facilitated by superfast sensorimotor responses to guide echo-based captures (*Stidsholt et al., 2021*). But these superfast movements may be less beneficial when gleaning since ground-dwelling arthropods can seek refuge under leaves or twigs if the attack of the bat is not perfectly aimed. In such cases, the bats must rely on tactile and olfactory cues, and their poorer ground locomotion to find the prey (*Kolb, 1958*), contributing to the low success ratio of gleaning (*Supplementary file 1*).

Despite the disparity in success ratios, tagged bats used both foraging strategies to capture food (*Figure 1*). During one night of foraging, the bats on average caught a mean of 25 insects in the air and 29 insects on the ground (*Figure 1*) demonstrating a reliance on both food sources. These numbers are on par with total prey captures of the similar-sized *Rhinopoma microphyllum*, estimated from buzz counts (*Cvikel et al., 2015*), but well below the very high feeding rates inferred in a smaller (7–11 g) species (*M. daubentonii*) (*Encarnação and Dietz, 2006*). The extra weight of the tags (~3–4 g) (*Portugal et al., 2018*; *Kline et al., 2021*) did not appear to strongly impact the number of prey attacks nor the ability to capture food since (i) both tagged (*Videos 1 and 5*) and un-tagged trained bats quickly learned to intercept aerial and ground prey in the lab with similar success ratios as in the wild (*Supplementary file 1*), and (ii) the wild tagged bats spent the same amount of time on foraging outside the colony as bats equipped with lighter (0.4 g) telemetry transmitters (*Egert-Berg et al., 2018*). The discrepancy between the inferred intake of thousands of insects per night from a smaller bat species (*Encarnação and Dietz, 2006*), and the measured total prey captures in our study is, therefore, more likely to relate to the dramatic difference in prey sizes between the prey species rather than a reduced foraging effort due to tagging effects.

Given our measured total prey captures and prey sizes, wild greater mouse-eared bats in our study assimilated an average of 61.4 kJ/night per bat (assuming 82% assimilation) (*Figure 3G*). This is higher than the estimated field metabolic rate (FMR) of the similar-sized female lesser long-nosed

bats (*Leptonycteris yerbabuenae*) of 40 kJ/day (*Goldshtein et al., 2020*), but close to the allometric scaling estimate of the FMR of a 30 g bat based on heart rate measurements from wild-tagged 18 g *Uroderma bilobatum* ($FMR_{M.myo} = FMR_{U.bil} * (30\ g/18\ g)^{0.7} = 65.5$ kJ/day) (*O'Mara et al., 2017*). Thus, despite making less than 100 prey captures per night, the estimated food intake of the tagged bats matches their predicted FMR. The bats in our study on average reached their predicted energetic requirement in a full night of foraging, but only just so, suggesting either feeding to satiation or perhaps alternatively that they might have little scope to compensate for changes in their environment. Since the bats fly out just after sunset and return early in the morning, they are vulnerable to any disturbance or change in habitat quality that reduces their foraging intake. Moreover, despite selecting only heavy, post-lactating females within the same colony, we measured a wide individual variation in hunting tactics. Tagged bats attacked from 48 to 280 prey during a night of foraging (*Figures 1 and 2*), demonstrating that continuous recordings from the same individuals are important to quantify the hunting efforts and energy budgets of wild predators, and that extrapolations of food intake from brief observations may result in significant errors in that light.

## Greater mouse-eared bats prefer larger ground-dwelling prey over aerial prey despite low success ratios

The tagged bats attacked prey in the air and on the ground at similar rates, but success ratios for aerial prey were more than twice of those of ground prey (*Figure 1F*). Moreover, gleaning insects most likely exposes bats to a higher predation risks from ground predators, and a higher risk of injury ( *Brandmayr, 2009*). This begs the question of why most of the tagged bats (N=22 of 34 bats) nonetheless preferred to capture insects on the ground? To address this, we hypothesized that bats would choose the foraging strategy with the highest profitability (i.e. energy intake/time) (*Stephens and Krebs, 1986*). We estimated prey profitability by dividing estimated prey caloric values with prey search and handling time, factoring in the success ratios for each strategy. We find that gleaning prey off the ground, despite the much higher failure ratios, renders a prey profitability between 2.5 and 14 times higher than aerial hawking owing to the much larger ground prey (*Figure 3F*). Beetles gleaned off the ground are 3–20 times heavier than aerial prey and have higher protein and fat content (*Razeng and Watson, 2015*). In fact, 85% of the energy returns from a night of foraging comes from gleaning despite the similar average number of prey attacks in the air and on the ground per night. Thus, the paradox of preferring low success ratios and higher risk of injury while gleaning is explained by a much higher prey profitability: the bats opt for a high risk-high gain strategy. Nevertheless, we find that aerial hunting remains as a valuable supportive foraging strategy for 19 out of 34 individuals on the nights sampled (*Figure 1DE*), contributing 15% of the total energy intake. This use of aerial hawking next prompted us to investigate how habitat affected prey profitability and the choice of foraging strategy.

## Prey profitability changes with habitat, but does not affect prey switching

We tested the hypothesis that bats maximize their energy intake by adapting their foraging strategies to the habitat (i.e. open fields vs forests *Arlettaz, 1999*, *Figure 2GH*) or movement style (i.e. commuting vs active searching for prey, *Figure 2—figure supplement 1*) (N=7 GPS-tagged bats). Both factors could affect the profitability of prey and thereby influence foraging decisions. The bats primarily gleaned prey during dedicated foraging bouts, whereas hawking was used equally during commute and in foraging bouts (*Figure 2—figure supplement 1*). This indicates that aerial hawking is a flexible strategy that is efficient enough to use during commuting. This perhaps offsets some of the energetic costs of the often lengthy commutes made by greater mouse-eared bats to preferred foraging areas while also potentially providing them with concurrent information about aerial prey profitability.

When actively gleaning for prey, the tagged bats were more successful and attacked more prey per bout in open fields than in forest habitats (26 vs 15 attacks/bout), but performed more foraging bouts in the forest (field: eight foraging bouts vs forest: 15 bouts; three bouts were a mixture between field and forest). Thus, the quality of the gleaning habitat (here defined as a habitat enabling higher success ratios and attack rates for the same strategy) does not appear to affect the decision of which habitat to target. This could have several explanations: bats may be maximizing energy intake by switching to

forest habitats after depletion of open field habitats, or they could be balancing predation risk and/ or conspecific competition which are both likely to be higher in open fields. In contrast to gleaning, bats were equally successful when using echolocation to capture insects above open fields and below the canopy in forests showing that these bats are well able to hunt in semi-cluttered spaces (*Stidsholt et al., 2021*). Such a flexible strategy allows them to be efficient hunters across diverse environments.

## Prey switching is adapted to changes in the environment

Although greater mouse-eared bats hunt independently, the dominant foraging strategies of individuals tagged on the same night were more similar than those for bats tagged on different nights. This indicates that temporally-varying environmental parameters such as rain, wind, or mass emergence of aerial insects, experienced by all bats in the area on a given night, influence the most beneficial foraging strategy. For example, after rain, the rustling sounds of walking arthropods on leaves are more difficult for bats to detect potentially halving the detection range (*Goerlitz et al., 2008*). Reduced detection ranges may not only lower detection rates but also capture success ratios (i.e. less time to plan and execute a capture). The time between gleaning prey attacks increased by an average of 35 s on the two nights with the most aerial captures (July 22, 2018 and 2019, GLMM; testing whether the time between gleaning attacks was explained by nights with a majority of aerial captures, $p<0.01$) and the bats switched more often between gleaning and aerial strategies on an attack-to-attack basis (GLMM; testing if the number of switches per night was explained by the nights with a majority of aerial captures, $p<0.01$). Longer periods between prey attacks and more strategy switching indicate that gleaning prey on these nights was less profitable compared to aerial prey, either due to poor conditions on the ground or a mass emergence of aerial prey.

Our results demonstrate that greater mouse-eared bats often resort to aerial hawking despite their preference and specialization for ground gleaning. Although much smaller prey are taken on the wing, the consistently high success ratios and prey capture rates of hawking independent of habitat make this a reliable backup strategy. The same qualities may make hawking a more widespread strategy than expected in other gleaning bat species. Such foraging flexibility both adds to the energy intake of wild bats and also improves their resilience to changing conditions. Thus, this strategy might have allowed a wide range of bat species to tap into the unpredictable, ephemeral food resources in open spaces. This would have put an evolutionary pressure on maintaining aerial hawking capabilities to secure an additional food resource in fluctuating environments, and may also explain why hawking remains as a foraging strategy in many bat species traditionally seen as gleaning and trawling specialists.

## Conclusion

We show that greater mouse-eared bats, a gleaning specialist, catch relatively few prey items per night on the ground at high failure ratios, but still achieve a high prey profitability by targeting large, energy-rich prey. This shows that prey availability and size, weighted by relative hunting success, are important drivers of foraging decisions in wild predators. We find that the bats use gleaning as a primary foraging tactic in a high risk-high gain approach, and switch to aerial hunting when environmental changes reduce the profitability of ground prey. We conclude that prey switching matched to environmental dynamics plays a key role in covering the energy intake even in specialized predators.

## Materials and methods

### Resource availability

Lead contact

Further information and requests for resources should be directed to and will be fulfilled by the lead contact, Laura Stidsholt (laura.stidsholt@bio.au.dk).

Materials availability

This study did not generate new unique reagents.

### Method details

All experiments were carried out under the licenses: 721/12.06.2017, 180/07.08.2018, and 795/17.05.2019 from MOEW-Sofia and RIOSV-Ruse. We tagged and recaptured 34 female,

post-lactating greater mouse-eared bats with sound-and-movement tags from late July to mid-August in the seasons 2017, 2018, and 2019. The bats were caught with a harp trap at Orlova Chuka cave, close to Ruse, NE-Bulgaria, in the early mornings as they returned to the roost. The bats were kept at the Siemer's Bat Research Station in Tabachka to measure the forearm lengths, CM3, and body weights (*Supplementary file 1*). Bats weighing above 29 g were tagged and released the either during the day in the cave or the following night between 10–11 p.m. at a field 8 km from the roost (Decimal degrees: 43.622097, 25.864917) or in the colony. The tags were wrapped in balloons for protection and glued to the fur on the back between the shoulders with skin bond latex glue (Ostobond). The bats on average spend one to eight days equipped with the tags until we recaptured the bats at the cave or the tags detached from the bats and fell to the ground below the colony. Upon recapture, the bats were weighed and checked for any sign of discomfort from the tagging before they were released back to the colony.

## Tags

We used two different tags for this study. Both tags recorded continuous data during one night of foraging. The first tag (Tag A) recorded audio data with an ultrasonic Knowles microphone (FG-3329) at a sampling rate of 187.5 kHz, with 16-bit resolution, a 10 kHz 1-pole analog high-pass filter, and a clipping level of 121 dB re 20µPa pk. These tags also included triaxial accelerometers sampling the movement of the bats at 1000 Hz with a clipping level of 8 g. All accelerometer data were calibrated, converted into acceleration units (m/s$^2$), and decimated to 100 Hz. The orientation of the bats was recorded with triaxial magnetometers using a sampling rate of 50 Hz. These tags weighed from 3.5 to 3.9 g (including VHF for localization and recapture of the bats). The second tag (Tag B) recorded audio using an ultrasonic Knowles microphone (SPU0410LR5H-QB) sampling at 94 kHz and with 16-bit resolution. The movement of the bats was recorded with triaxial accelerometers at a 50 Hz sampling rate and with a clipping level of 8 g. These tags also included GPS sensors that logged the position of the bats every 15 s. These tags weighed 3.9–4.2 g (including VHF). In total, we have data from 16 bats with Tag A and 18 bats with Tag B (of which five are recording at a 50% duty ratio meaning that the tags record for 30 s and is turned off for 30 s to save battery and prolong recording time). Of the 18 tags B recovered, seven tags had GPS data in addition to the accelerometer and audio data (*Supplementary file 1*).

## Tagging effects

Both tags weighed 11–15% of the body weight of the tagged bats. We addressed the effects of tagging on the data and the bats by the following procedures: (i) We trained bats in a flight room to capture mealworms and moths tethered on strings and glean beetles of different sizes from either a bowl or a square meter of natural forest floor (*Supplementary file 1*). The bats caught aerial prey with high success ratios (95% for mealworms and 69% for moths (N=2 bats)) with tags and without tags (75% for mealworms (N=4 bats)). The bats gleaned beetles with a success ratio of 39% from the square of the forest floor. The success ratio increased to 85% when they were catching beetles from a bowl with no escape options for the prey (N=3 bats). This indicates that the low gleaning success ratios in the wild are most likely not caused by tagging effects, but more likely because beetles can escape in cracks or below leaves. Additionally, we could not detect any visual difficulties with capturing prey from strings or beetles from the ground. (ii) The wild bats caught prey with high success ratios in the air indicating that the tag effects had little or no impact on the ability to capture prey (*Videos 1 and 5*). (iii) A previous study with the same species and tags found that tagged and untagged bats spend an equal amount of time foraging (*Egert-Berg et al., 2018*). (iv) The weight loss of these bats of 7% during the instrumentation time was equal to the weight loss of control bats carrying only VHF radio transmitters (0.5 g) indicating that handling and carrying of tags might disturb the bats, but the additional extra load did not seem to add further energetic consequences to the bats in addition to the VHF (*Egert-Berg et al., 2018*).

## Quantification and statistical analysis

To understand how the tagged bats allocated their time and captured prey in the wild, all wild tag recordings were manually analyzed by displaying the acoustic and the movement data in 7–20 second segments with an additional option of playing back audio data. The visualization included three

separate windows with synchronized data: (i) An envelope of audio data filtered by a 20 kHz four pole high pass filter to detect the echolocation calls. (ii) A spectrogram of audio data filtered by a 1 kHz one pole high pass filter to visualize the full bandwidth acoustic scene showing echolocation calls, conspecific calls, chewing sounds, wind noise, etc. (iii) The final window showed triaxial accelerometer aiding the identification of wingbeats, landing, take-offs as well as capture events.

## Categorization of prey attacks

Greater mouse-eared bats are known to glean prey off the ground and to capture aerial prey. To recognize gleaning prey attacks in the wild data, the tag recordings were ground-truthed by analyzing sound and movement data from prey attacks of bats under controlled experimental settings in the lab. Two individuals were trained to catch walking beetles on vegetation using passive hearing while carrying a tag. These ground captures were identified by stereotyped patterns consisting of three simultaneous events (i) Low vocalizations (around 50 dB re 20μPa$^2$s) prior to the prey attack indicating that the bat was using passive listening to listen for prey-generated cues. (ii) A short, broadband, and loud audio transient simultaneous to a peak in the accelerometer data indicating that the bat was landing on the ground. (iii) The accelerometer signal indicating a landing was an increase in wingbeat frequency and amplitude prior to the landing, a peak in the sway and heave axis (y and z dimensions, often with opposite values) at the time of contact with the ground and often followed by a flattening of the signal on all three axes. These stereotyped audio and accelerometer signals found in the laboratory experiments were matching signals seen in the wild data (*Figure 1—figure supplement 1*). These signals were then identified in all wild tag recordings during the visualization and marking process. Aerial prey attacks were identified in the wild data if a buzz was present. Only buzzes in flight were marked to exclude landing buzzes. In addition, each prey attack was marked as 'successful' or 'unsuccessful' based on the presence or absence of chewing sounds. The chewing sounds were audible (*Video 6*) and for the low-noise tag recordings were visible in spectrograms (*Figure 3—figure supplement 1*). For the five 50% duty cycled tag recordings we doubled the foraging attempts and successes. The manual inspections of the capture attacks were verified by comparing the results for aerial captures with an automatic buzz-detector that detected when more than six consecutive call intervals were below 8 ms. We excluded all buzzes emitted prior to landing since landing buzzes are not associated with prey attacks. The comparison between the automatic and manual identifications of prey attacks shows that the manual inspection of the attacks is reliable within 11% (quartiles: 4 to 17%) (*Figure 1—figure supplement 2*). Furthermore, the average time difference between the automatically and manually detected attacks was 0.9 s (quartiles: 0.68–1.31 s) and, therefore, verifies the method of manual inspection of the data (*Figure 1—figure supplement 2*).

## Behavioral analysis (Tag type A)

To evaluate time allocation, we analyzed data from 15 tags (tag type A) that included both accelerometer and magnetometer data necessary for the analysis. We separated times of rest from flight through the identification of wingbeat epochs. Wingbeats were detected as cyclic oscillations in the z-axis dimension of the accelerometer data. We first band-pass filtered the z-axis dimension of accelerometer data (from 5 to 25 Hz) by a delay-free symmetric FIR filter (filter length: 1024 samples, sample rate: 100 Hz). We then identified flight epochs as the time intervals where a running mean of 50 s of the wingbeat data was above a threshold of 20 m/s$^2$. A window length of 50 s was chosen to avoid short, flight epochs consisting of only a few wingbeats.

For each flight epoch, we identified times of foraging from traveling by changes in the heading because the bats fly straight towards and between foraging grounds (*Egert-Berg et al., 2018*). When the bats fly straight to the foraging grounds during commuting flights, the heading is stable over time since they maintain a fixed straight body position during flight. However, when foraging at hunting grounds, the bats fly around in circles, loops or perform erratic flight maneuvers that are translated into a higher variation in the heading compared to when bats fly straight. We used this difference in flight pattern to separate comuting and foraging flights. To do this, we first computed heading over the entire night by gimballing low-pass (3 Hz) filtered triaxial magnetic field measurements with the pitch and roll estimated from down-sampled and low-pass filtered (3 Hz) accelerometer data (*Johnson and Tyack, 2003*). We applied a running mean of 50 s to the heading measurements to evaluate foraging bouts. We chose a length of 50 s similar to the minimum foraging bout length used by *Hurme et al.,*

*2019* and in our GPS analysis. Foraging bouts were identified as time intervals where the envelope of the signal was above a threshold of 0.05. Due to the large variation between animals, this threshold was raised to 0.3 for six tag recordings to avoid more than 10 switches between traveling and foraging per night as used by *Hurme et al., 2019* and in our GPS analysis. We omitted identified foraging bouts with no prey attacks from the analysis (N=62 out of 202 foraging bouts for 16 bats).

## GPS and habitat analysis (Tag type B)

We recaptured seven tags with GPS data. We used first-passage time analysis (*Fauchald and Tveraa, 2003*) to identify foraging bouts from traveling bouts (*Hurme et al., 2019*) by using the R package 'adeHabitatLT' (*Calenge, 2006*). First, we converted the latitude and longitude coordinates from degrees to meters ('proj4string' function in adeHabitatLT) and then regularised the GPS tracks to 10 s time stamps. We then calculated the first-passage times for all radii ranging from 5 to 400 m (in 5 m steps). We plotted the variance of the log-transformed first-passage times and found the highest value of around 250 m for all bats which is in accordance with a previous study on *Myotis vivesi* (*Hurme et al., 2019*). This value estimates the scale at which the bat is operating, and was used for all tracks. We used the Lavielle method ('lavielle' in adeHabitatLT) to divide the path segments into foraging and non-foraging bouts (*Calenge, 2006*). The minimum number of locations in a bout was chosen to be five (corresponding to 50 s), and the maximum number of segments per night was 50. The function 'chooseseg' was chosen to find the number of segments at which the contrast between bouts was highest. The number of segments per night was estimated to either be 10 or 11 per bat. To find a threshold value to separate foraging and traveling bouts, we found that the distance traveled per segment showed bimodal distribution (function 'bimodalitycoeff' in Matlab *Zhivomirov, 2022*). We then fitted a Gaussian mixture model with two components to the data. We defined the threshold between the two distributions as the lowest quartile of the non-foraging (i.e. traveling) segments corresponding to 40 meters traveled per 10 s segment.

To determine in which habitats the bats were foraging, we transferred all tracks onto a Google Map Satellite Imagine (using: 'plot_google_map' in Matlab version 2021b). We manually determined whether the GPS locations for each segment were located in one of three categories: Field, Forest, or both/others. We excluded foraging (N=4) and non-foraging (N=23) bouts that were not assigned to either field or forest.

## Statistical analysis

All statistical modeling was performed in R (version 4.0.3). We fitted different models to understand how prey attacks and success ratios were influenced by the foraging strategy, the habitat, and the mode of action (commuting vs foraging in bouts). For all models, we used a goodness of fit evaluation based on the marginal ($R_{(m)}^2$) and the conditional R2 ($R_{(c)}^2$) (Nakagawa and Schielzeth, 2013), and Bat ID as a random effect.

Model 1: We investigated how prey attacks and foraging successes were influenced by the foraging strategy (N=34) and tagging effects. We used foraging strategy, tag type, tag percent weight, and bat lost body weight as predictor variables and fitted two linear mixed effect models to the data ('lme4' R-package *Bates, 2015*). In the first model (1 a), we used foraging attempts as a response variable assuming a Poisson distribution (link='log') after investigating the distribution via histograms, and in the second model (1b), we used foraging success ratios as a normally distributed response variable. We examined potential collinearity between these predictor variables using variance inflation factors ('vif' in R from package 'car' *Fox and Weisberg, 2019*). There was collinearity between tag type and tag weight, so we excluded tag type from the model. We examined the residuals ('DHARMa' package in R) and found deviations from the expected distribution for model 1 a. We, therefore, refitted the model with a gaussian distribution which improved the residuals. Only slight deviations from the expected residuals distribution was found for model 1b. Model 1 a showed that foraging strategy explained 15% of the amount of prey attacks, and that double the amount of prey was attacked when gleaning (*Supplementary file 1*). Model 1b revealed that bats are more successful when hawking for prey (*Supplementary file 1*, p<0.00001) and that foraging strategy explains 74% of the deviance in success ratio. In both models, the differences in tagweight in percent of the bat body weight did not have any effect on the response variable (prey attakcs, p=0.87 and success ratio p=0.9). Since p-values

could not be extracted in R, p-values were added from the same analysis in Matlab (mathworks, version 2021b).

Model 2: We investigated how the prey attacks and foraging successes were influenced by the habitats of the seven bats tagged with GPS tags (Tag B). For both models, we used three categorical predictor variables: habitat type (field vs forest), foraging strategy (aerial vs hawking), and movement style (commuting vs foraging in bouts).

In model 2 a, we used prey attacks (sum of failed and successful prey attacks in each foraging bout) as a response variable and fitted a GLMM ('glmer' function in R-package 'lme4') to the data with a Poisson distribution of the response variable and a 'log' link function. In model 2b, we used foraging success (per foraging bout) as a normally distributed response variable and fitted a LMM to the data ('lme' function in R-package 'nlme' *Pinheiro, 2022*). We used model selection procedures ('dredge' in R-package *Burnham and Anderson, 2002*) to examine the best-fitted models using the AICc (corrected Akaike information criterion). The best-fitted models included all three predictor variables in both models 2 a and 2b. We examined the residuals ('DHARMa' package in R) and slight deviations from the expected distributions were found.

In model 2 a, the three predictor variables explained 45.0% of the data, while the random effects explained 52% of the deviance in prey attacks. Thus, model 2 a revealed that the variance in prey attacks per night is more explained by individual bats rather than habitat, strategy or movement style (*Supplementary file 1*). In model 2b, the three predictor variables explained 72.3% of the deviance of the success ratios. No difference was found when subtracting the random effect for model 2b. This model revealed that there is strong evidence that foraging strategy affects the success ratio of the bats (*Supplementary file 1*), and that habitat has a weak effect on the success ratio. Foraging style does not have an effect on the success ratio.

Model 3: We tested whether the immediate environment affected the foraging decision in bats by investigating the relationship between the dominant foraging strategy and the night of tagging for all 34 bats. The dominant foraging strategy was found as the ratio between aerial and foraging attempts ranging from 0 to 1 (0 meaning 100% aerial foraging; 1 meaning 100% gleaning) and was used as a normally-distributed response variable. The data was fitted with a LMM (function: 'lmer' in R) using release night as a categorical fixed effect, and release site as a categorical random effect. Our first model included tag type, tag weight in percent of the bat weight, and lost weight during instrumentation as fixed effects. We examined potential collinearity between these predictor variables using variance inflation factors ('vif' in R from package 'car' *Fox and Weisberg, 2019*). There was collinearity between tag type and tag weight, so we excluded tag type from the model. We used model selection procedures (dredge in R-package 'MuMIn' *Burnham and Anderson, 2002*) to examine the best-fitted model using the AICc criterion. The best-fitted model included the night of tagging and release site as the explanatory variable, but excluded tag weights. This suggests that the bats chose a foraging strategy independent on the weight of the tag. The best-fitted model showed that there was strong evidence of individual night on the chosen foraging strategy, indicating that bats on the same nights choose the same foraging strategy (N=10 different nights, 34 bats, and a mean of 3 (2.1 SD) tags per night (*Figure 1—figure supplement 3*, *Supplementary file 1*)).

Out of 34 retrieved tags in this study, 24 tags were retrieved by recapturing the bats (compared to finding the tags on the cave floor). We weighed the bats after detaching the tags and calculated the change in body weight before or after instrumentation in percent. We investigated the change in body weight during the instrumentation might affect the dominant foraging strategy, the prey attacks, and the success ratio. We fitted linear models to the data ('lm' in R) and calculated the independent contribution of each predictor variable (using 'hier.part' in R *Nally and Walsh, 2004*). The change in body weight did not have an effect on the foraging strategy, success ratio, or number of prey attacks since the other predictor variables explained 69%, 62%, and 65% of the variance in the data and all p-values for the tag predictor variables were un-significant. This suggests that the bats chose foraging strategies independent of the effects of tags despite high tagging weights of 10–15% of the body weight of the bats.

## Relative prey sizes based on quantification of mastication sounds

After the visualization and marking of all prey attacks, a custom-written chewing detector was used to automatically identify mastication sounds for nine tag recordings with sound quality to perform this

analysis. This analysis included 452 ground capture attempts and 387 aerial prey attacks. The detector was used in two steps: (i) To automatically determine whether the prey attack was successful or not to verify the manual decision process based on listening to the chewing sounds. (ii) To characterize the mastication sounds of each prey item from successful captures.

We extracted mastication sounds when the bat was flying after each prey attack. Since the bats in flight chewed between echolocation calls, we analyzed the intercall interval from the time of capture to either the next prey attack or 100 s ahead. We first detected mastication sounds and then classified the mastication sounds for each intercall interval.

Each intercall interval was extracted, filtered by a 7–15 kHz 4-pole Butterworth filter, and convolved with a 40 ms Hanning window to exclude transients. These parameters were based on mastication sounds from capture bats with peak frequencies of 7 kHz which corresponds to the same peak frequency in wild bats. The intercall interval was classified as containing a mastication sound if the maximum amplitudes of the filtered signals were above a threshold of 0.012 and the peak frequency was above 5 kHz and below 20 kHz (*Figure 1—figure supplement 4*, black vs red). The intercall intervals that included mastication sounds were then used for the classification. Here, we filtered the intercall intervals of the original sound data containing mastication with a 5 kHz 4-pole high-pass filter to reduce flow noise. To extract the onset of chewing as well as the duration, the detector also automatically extracted the time at which the bat produced the 10th and 90th quantiles of the chewing sounds (*Figure 1—figure supplement 4*, gray dashed lines). The 10th and 90th quantiles were a conservative choice to avoid false detections. The onset of the chewing was determined when the bat emitted the first sound in this interval. The duration of the chewing was determined as the length of this interval. The handling time was estimated as the sum of the onset and duration of the mastication.

To test the performance of the classifier, the automatic and manual classification of successful vs unsuccessful prey captures were compared. A confusion matrix was made separately for all ground and aerial prey attacks. The classifier was evaluated by calculating the positive predictive value (PPV) and the false negative rate (FNR) based on the derivatives of the confusion matrix from the classification of the ground and aerial prey attacks (*Supplementary file 1*). Overall, the detector worked with PPV values above 0.99 (*Supplementary file 1*). However, the FNR was higher for aerial captures (3% for ground; 9% for aerial captures). The maximum sound energy in chewing after aerial captures were weaker than after ground captures which may explain the worse performance in the air. To verify the mastication detector, we extracted and characterized mastication sounds of known prey types and sizes in controlled setups in the laboratory where the bats were eating while flying after prey captures (*Figure 3—figure supplement 1*). We performed these experiments by tethering either small or large moths on strings in a flight room or releasing small mealworm beetles or larger cockroaches on the floor. We then led the bats to capture prey freely while carrying tags and noted what size and type of prey the bat caught at any given time. For the analysis, we then extracted all captures and chewing durations based on the same method as for the wild data. We caught the moths in light traps at night outside the flight rooms, but were unable to catch enough ground beetles in pitfall traps for our experiments. Instead, we fed the bats mealworm beetles and cockroaches since these beetles were easier to get access.

## DNA metabarcoding

To understand the taxonomic diversity of the diet of wild mouse-eared bats we collected fecal samples from 54 bats (n=26 in 2017, n=28 in 2019, n=48 female bats) returning to the roost after foraging. Individual bats were caught with a harp trap positioned at the entrance of Orlova chucka cave, Pepelina, Bulgaria, and placed in individual clean cotton bags until defecating. Fecal samples were collected in 98% alcohol and stored until further analysis. DNA extraction, data sequencing, and bioinformatics were done following *Morinière et al., 2016* (see also ; *Morinière et al., 2019*). In short, DNA from the fecal samples was extracted by using the DNEasy blood & tissue kit (Qiagen) following the manufacturer's instructions. Multiplex PCR was performed using 5 µL of extracted genomic DNA and high-throughput sequencing (HTS)-adapted mini-barcode primers targeting the mitochondrial CO1 region. HTS was performed on an Illumina MiSeq v2 (Illumina Inc, San Diego, USA) at AIM - Advanced Identification Methods GmbH, Leipzig, Germany. Further, FASTQ files were combined and sequence processing was performed with the VSEARCH v2.4.3 suite (*Rognes et al., 2016*) and cutadapt v1.14 (*Martin, 2011*). Quality filtering was performed with the fastq_filter program of

VSEARCH, fastq_maxee 2; a minimum length of 100 bp was allowed. Sequences were dereplicated with derep_fulllength, first at the sample level and then concatenated into one fasta file, which was then dereplicated. Chimeric sequences were then detected and filtered out from the resulting file. The remaining sequences were clustered into OTUs (Operational Taxonomic Units) at 97% identity. To reduce likely false positives, a cleaning step was employed that excluded read counts in the OTU table of less than 0.01% of the total read number. OTUs were blasted against a custom Animalia database downloaded from BOLD (Barcode Of Life Database, http://www.v3.boldsystems.org) and BIN (Barcode Index Number) information.

We measured the lengths of representatives of each species from online photo databases (http://www.boldsystems.org) or the reference collection of The National Museum of Natural History Sofia (Bulgaria) covering the same region in Bulgaria. The lengths used for the calculations were estimated as the maximum values of the measured prey lengths.

### Caloric value estimations

To estimate the energetic intake of one night per bat, we first used length-weight regressions to convert arthropod body lengths to body masses (*Straus and Avilés, 2018*). Here, we used the weighted average of the two primary arthropod orders from each foraging strategy: ground diet (i.e. Carabidae (78%) and Orthoptera (22%)) and aerial diet (i.e. Diptera (65%) and Lepidoptera (35%)) (*Straus and Avilés, 2018*). From this calculation, we estimated the dry body masses of ground and aerial prey only based on metabarcoding data. In addition to this, we also calculated the dry body mass of aerial prey based on the ratio between the mastication (number of masticatations/capture) after ground and aerial prey. This gave us an additional value for the dry body mass of aerial prey. Thus, we proceeded with a higher and lower estimate of aerial prey dry masses in the following calculations.

To convert dry body masses (mg/prey) into caloric values (J/prey), we multiplied the dry body masses of the prey with the caloric values for either ground (*Bell, 1990*; *Zygmunt et al., 2006*) or aerial (*Kurta and Kunz, 1987*; *Bell, 1990*) prey. Here, we used the caloric values of the weighted average of the two most numerous prey orders for ground and aerial prey. Nightly caloric intake of each bat was then calculated by multiplying the number of eaten ground and aerial prey items with the caloric values of each prey type.

### Profitability index calculations

To compare the profitability between the two foraging strategies, we calculated the relative profitability of each prey using the following relationship:

$$P(i) = \frac{Preycaloricvalue(i)}{(Handlingtime(i) + Timebetweenpreyattacks(i))} * Successratio(i),$$

$i$=foraging strategy, Success ratio (Mean success ratio per bat (per bout with more than one capture) as a fraction from 0 to 1), Time between prey attacks (mean number of seconds between prey attacks per bat per night with unit seconds). Since the wingbeat strength and wingbeat frequency was the same between prey attacks when gleaning and hawking (*Figure 3—figure supplement 3*), we used the time between attacks as a proxy for hunting effort estimated as the time between attacks. Prey caloric values were estimated using both metabarcoding and mastication analysis (see *Caloric value estimations*). The prey profitability unit is, therefore, J/s per prey capture.

## Acknowledgements

We are thankful to Kaloyana Kosseva for help with bat feces collection in the field and to Ilias Foskolos and Nor Amira Abdul Rahman for assistance with laboratory experiments. We are grateful to the entire crew at the Siemers Bat Research Station for the support during the seasons 2017–2019 and to the Directorate of the Rusenski Lom Nature Park, Bulgaria.

## Additional information

### Funding

| Funder | Grant reference number | Author |
|---|---|---|
| Villum Fonden | 41386 | Laura Stidsholt |
| Bulgarian Academy of Sciences | DFNP-17-71/28.07.2017 | Antoniya Hubancheva |
| Carlsbergfondet | Semper Ardens | Peter T Madsen |
| Deutsche Forschungsgemeinschaft | 241711556 | Holger R Goerlitz |

The funders had no role in study design, data collection and interpretation, or the decision to submit the work for publication.

### Author contributions

Laura Stidsholt, Conceptualization, Data curation, Formal analysis, Investigation, Visualization, Methodology, Writing – original draft, Project administration, Writing – review and editing; Antoniya Hubancheva, Data curation, Formal analysis, Visualization, Methodology, Project administration, Writing – review and editing; Stefan Greif, Conceptualization, Supervision, Investigation, Methodology, Project administration; Holger R Goerlitz, Conceptualization, Resources, Supervision, Funding acquisition, Investigation, Project administration, Writing – review and editing; Mark Johnson, Software, Formal analysis, Supervision, Investigation, Methodology, Writing – review and editing; Yossi Yovel, Conceptualization, Resources, Funding acquisition, Investigation, Methodology, Writing – review and editing; Peter T Madsen, Conceptualization, Resources, Supervision, Investigation, Methodology, Project administration, Writing – review and editing

### Author ORCIDs

Laura Stidsholt ⓘ http://orcid.org/0000-0002-2187-7835
Antoniya Hubancheva ⓘ http://orcid.org/0000-0001-8362-1301
Peter T Madsen ⓘ http://orcid.org/0000-0002-5208-5259

### Ethics

All experiments were performed under the licenses: 721/12.06.2017, 180/07.08.2018 and 795/17.05.2019 from MOEW-Sofia and RIOSV-Ruse. Every effort was made to minimize suffering.

### Decision letter and Author response

Decision letter https://doi.org/10.7554/eLife.84190.sa1
Author response https://doi.org/10.7554/eLife.84190.sa2

## Additional files

### Supplementary files
- MDAR checklist
- Supplementary file 1. Metadata on bats, tagging experiments and statistical models.

### Data availability

Data and code available at the Mendeley Data Respository: https://doi.org/10.17632/smx7n93syw.1.

The following dataset was generated:

| Author(s) | Year | Dataset title | Dataset URL | Database and Identifier |
|---|---|---|---|---|
| Stidsholt L, Hubancheva A, Greif S, Goerlitz HR, Johnson M, Yovel Y, Madsen PT | 2022 | Data from: Echolocating bats prefer a high risk-high gain foraging strategy to increase prey profitability | https://data.mendeley.com/datasets/smx7n93syw | Mendeley Data, 10.17632/smx7n93syw.1 |

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
