## [Editor Report]

This study presents important findings on the hunting strategies and energy intake of bats in the wild. It combines several methods (biologging, captive experiment, and DNA metabarcoding) to provide convincing evidence for the claims. With detailed data and analyses on foraging ecology, this work will be of broad interest to animal ecologists.

---

## [Decision Letter]

**Decision letter after peer review:**

Thank you for submitting your article "Echolocating bats prefer a high risk-high gain foraging strategy to increase prey profitability" for consideration by *eLife*. Your article has been reviewed by 3 peer reviewers, including Yuuki Y Watanabe as Reviewing Editor and Reviewer #1, and the evaluation has been overseen by a Reviewing Editor and Detlef Weigel as the Senior Editor. The following individual involved in the review of your submission has agreed to reveal their identity: Brock Fenton (Reviewer #2).

Essential revisions:

All reviewers are generally positive about the manuscript. However, reviewers 1 and 2 commented that the manuscript could be framed better, especially in introduction and discussion, so that more general readers in the field of animal ecology appreciate its significance (reviewer 1). Related to this, clearly articulated hypotheses are needed (reviewer 2). In addition, many specific comments raised by reviewer 3 should be addressed.

*Reviewer #1 (Recommendations for the authors):*

The authors set several working hypotheses in the Introduction, but these are quite specific to the study species and it's unclear to me why these hypotheses need to be tested in the first place. Results and discussion could be written more concisely. In the current form, it's hard to follow the mainstream and capture the main message of the manuscript (I think this is partly because the working topic is not well introduced in the Introduction). I feel that the manuscript needs to be reconstructed so that it can appeal to a broader, animal ecology community (the beginning sentences in the Discussion are good examples). One option the authors may consider is to reformat the manuscript to the standard research paper style (where results and discussion are well separated) rather than the current, letter style. The current format does not seem to be the best choice, because I had difficulty in following the stream.

*Reviewer #2 (Recommendations for the authors):*

To be clear, I think that the paper could be excellent if it were clearly focused and the predictions helped lead the reader (especially the nonspecialist reader) through the details.

The writing is very dense, which also makes a barrier for the reader. The lack of focus and predictions makes it difficult for the reader (at least this one) to appreciate the major strengths and weaknesses. The techniques and data are more than just fine, but the context is lacking.

*Reviewer #3 (Recommendations for the authors):*

Supplemental laboratory experiments are currently striving to demonstrate the accuracy of the proposed method to the extent possible, but there are a few points that are unclear or commented on, mainly concerning the experimental method and analysis. Below are some specific comments.

L63 "intervals of high variation in heading" Related to L430-442, it would be better to have a little more explanation (even in Method).

P91 successful (black edge) or failures (pink edge) yellow edge, black edge?

Figure 1DE As I commented in Table S1, TagB is heavier than Tag A, this may be the reason why gleaning is used more often in the foraging strategy.

Figure 2 As mentioned in the discussion, in addition to weather, what is the relationship between factors other than habitat in individuals that switch between gleaning and hawking relatively frequently and on foraging days? What about the relationship between a flight time from nest emergence and the number of food obtained up to that point (not only the actual number of successes but also the relationship with success rate)?

L174-188 Various profitability estimates are given here, but since these are expected to vary depending on the season and gender, it would be helpful to note that these values are for the July-August period of this study.

Table S1 Almost all bats showed a significant decrease in recapture weight from the values shown in the table. I suggest that the changes in body weight should be included in your statistical model to see if wearing a heavy tag that is nearly 15% of its own weight has any effect on its foraging behavior in the field. If we can say that there is no impact, that would be a very valuable indicator for future bat biologging studies.

L360 MEME microphone – Please write down the model number of the microphone.

P365 I think "50% duty cycled" means intermittent recording at a 50% duty ratio, but I think it needs a little more explanation so that new readers can understand it.

L376 "we could not detect any visual difficulties with capturing prey from strings or beetles from the ground (Video S1-2)" I thought this was to check if bats carrying tags have any difficulties when foraging, but the bats in the video are not carrying tags. Please explain further so that readers do not misunderstand.

L381 "The weight loss of these bats of 3-4%…" Table S1 shows that some individuals have a weight loss of nearly 10%. As I commented in Table S1, it is helpful to examine the effect of tagging in the lab, but it would be better to examine the effect of tagging on the actual field data.

L403 "Two individuals were trained to catch walking beetles on vegetation using passive hearing while carrying a tag (Video S1)" The video shows a close-up of a bat biting a beetle. It does not seem to be directly related to the explanation in the text here.

Figure S5 In this research, I thought that the accuracy of identifying the capture attempt here is probably one of the most critical factors. In this example, the pattern is similar to the one observed in the laboratory, but I don't know how reliable the results can be by visual identification. If possible, it would be good to have more supporting data (e.g. comparing labeling results between two persons).

L419 There is no such spectrogram in Figure 3.

L422 Are all of these 15 TAGs Tag A? Please describe any criteria you used to select these 15 tags.

L430-442 I thought it would be easier to understand how to determine the time of foraging if there was a diagram in the Supplement.

L444 L364 states that 18 Tag B loggers were recovered. Does this mean that the number of loggers recovered for Tag B was 18, of which 7 contained GPS data?

L524 "identify mastication sounds for nine tag…" The authors mentioned that masticatory sounds were identified for 9 tags, but in Figure 1D, E, and F, the percentage of success or failure is shown for all 34 loggers as data. I am wondering if I am misunderstanding something, but was the success/failure judgment made for all 34 logger data through the masticatory sound analysis?

L558 Table S9 is missing a title. The chewing sounds were measured in the lab on the prey of known species and size, but please explain the situation a little more.

---

## [Author Response]

Reviewer #1 (Recommendations for the authors):The authors set several working hypotheses in the Introduction, but these are quite specific to the study species and it's unclear to me why these hypotheses need to be tested in the first place. Results and discussion could be written more concisely. In the current form, it's hard to follow the mainstream and capture the main message of the manuscript (I think this is partly because the working topic is not well introduced in the Introduction). I feel that the manuscript needs to be reconstructed so that it can appeal to a broader, animal ecology community (the beginning sentences in the Discussion are good examples). One option the authors may consider is to reformat the manuscript to the standard research paper style (where results and discussion are well separated) rather than the current, letter style. The current format does not seem to be the best choice, because I had difficulty in following the stream.

We fully agree that the introduction was too specific for a general audience. We have, therefore, rewritten the introduction so that is starts with a much broader focus on animal ecology. The manuscript is formatted as a standard research paper style, but several sentences and paragraphs have been rewritten for clarity and to better capture the main message of the manuscript. Specifically, we have carefully outlined three hypotheses in the introduction that we test one by one throughout the discussion using subheadings to guide the reader.

Reviewer #2 (Recommendations for the authors):To be clear, I think that the paper could be excellent if it were clearly focused and the predictions helped lead the reader (especially the nonspecialist reader) through the details.The writing is very dense, which also makes a barrier for the reader. The lack of focus and predictions makes it difficult for the reader (at least this one) to appreciate the major strengths and weaknesses. The techniques and data are more than just fine, but the context is lacking.

Thank you. We agree that the writing in the original manuscript was dense offering a hard read for non-specialists. In the revised manuscript, we offer a broader start of the introduction with clear hypotheses, and we have added subheadings during the result and Discussion section to mirror these hypotheses and rewritten paragraphs of the discussion for better clarity.

Reviewer #3 (Recommendations for the authors):Supplemental laboratory experiments are currently striving to demonstrate the accuracy of the proposed method to the extent possible, but there are a few points that are unclear or commented on, mainly concerning the experimental method and analysis. Below are some specific comments.L63 "intervals of high variation in heading" Related to L430-442, it would be better to have a little more explanation (even in Method).L430-442 I thought it would be easier to understand how to determine the time of foraging if there was a diagram in the Supplement.

We have added more explanatory text (line: 80-83) to explain the difference between the tag types, and added an extra paragraph in the methods to explain the reasoning behind the method (line: 472-477). We agree that a diagram would be great to explain the analysis better, but with the current version of the manuscript, we are already at twelve supplemental figures, so we have not included further figures.

P91 successful (black edge) or failures (pink edge) yellow edge, black edge?L360 MEME microphone – Please write down the model number of the microphone.L419 There is no such spectrogram in Figure 3.P365 I think "50% duty cycled" means intermittent recording at a 50% duty ratio, but I think it needs a little more explanation so that new readers can understand it.L422 Are all of these 15 TAGs Tag A? Please describe any criteria you used to select these 15 tags.

Good suggestions that are all accommodated in the revised manuscript.

L558 Table S9 is missing a title. The chewing sounds were measured in the lab on the prey of known species and size, but please explain the situation a little more.

Good point: We have added a longer explanation of the experimental procedure in the method section (line: 634-641).

L174-188 Various profitability estimates are given here, but since these are expected to vary depending on the season and gender, it would be helpful to note that these values are for the July-August period of this study.

Good point, this has been added to the manuscript (line: 212).

L444 L364 states that 18 Tag B loggers were recovered. Does this mean that the number of loggers recovered for Tag B was 18, of which 7 contained GPS data?

Yes, this is correct. We have added an additional sentence to make this point clear (line: 400-403).

L524 "identify mastication sounds for nine tag…" The authors mentioned that masticatory sounds were identified for 9 tags, but in Figure 1D, E, and F, the percentage of success or failure is shown for all 34 loggers as data. I am wondering if I am misunderstanding something, but was the success/failure judgment made for all 34 logger data through the masticatory sound analysis?

Thank you for pointing out the lack of clarity in explanation of the results. We used the presence of mastication sounds to identify whether each prey attack was successful or not for all tag recordings (N = 34 bats). However, to make a more detailed qualitative analysis of chewing duration, we used only the nine tag recordings with the highest sound quality that would support this type of analysis. This has been clarified in the revised manuscript (line: 192-193).

L403 "Two individuals were trained to catch walking beetles on vegetation using passive hearing while carrying a tag (Video S1)" The video shows a close-up of a bat biting a beetle. It does not seem to be directly related to the explanation in the text here.

Thanks for pointing to this mistake, we have updated the text to exclude the video reference.

L376 "we could not detect any visual difficulties with capturing prey from strings or beetles from the ground (Video S1-2)" I thought this was to check if bats carrying tags have any difficulties when foraging, but the bats in the video are not carrying tags. Please explain further so that readers do not misunderstand.

Thank you for noticing this mistake. We have updated the video references accordingly.

Figure S5 In this research, I thought that the accuracy of identifying the capture attempt here is probably one of the most critical factors. In this example, the pattern is similar to the one observed in the laboratory, but I don't know how reliable the results can be by visual identification. If possible, it would be good to have more supporting data (e.g. comparing labeling results between two persons).

We agree with your point and have added an additional figure (Figure 1—figure supplement 2) to address this issue. Even though we agree that comparing labelled data between two people would be a good solution, we have instead chosen to compare our manual analysis with a reproducible automatic detection of aerial captures. We automatically identified buzzes in flight based on an automatic buzz-detector, and compared the results of the automatic detections and the manual detections. The automatic detector followed the trend of the manual detections with an average difference of 11 percent. We found a median time difference between automatic and manual detections of 0.9 s (90^th^ quartiles are 6.7 s). This finding is in strong support of the accuracy of identifying aerial captures manually and we have added this section to the methods (line: 458-465).

Table S1 Almost all bats showed a significant decrease in recapture weight from the values shown in the table. I suggest that the changes in body weight should be included in your statistical model to see if wearing a heavy tag that is nearly 15% of its own weight has any effect on its foraging behavior in the field. If we can say that there is no impact, that would be a very valuable indicator for future bat biologging studies.L381 "The weight loss of these bats of 3-4%…" Table S1 shows that some individuals have a weight loss of nearly 10%. As I commented in Table S1, it is helpful to examine the effect of tagging in the lab, but it would be better to examine the effect of tagging on the actual field data.

We completely agree with your comments and the importance of addressing any tagging effects. The bats on average lost 6.7 % of their body mass during the instrumentation. For some of the bats, we released them during the day and recaptured them at night. Since bats naturally lose weight across the day during fasting, a small part of the weight loss may be ascribed to the daily offset between release- and recapture weights. However, we agree that adding tag weights, tag weights according to bat body mass, and the change in body weight during instrumentation as explanatory variables in our models are important. We have added these parameters to our models, updated the tables, and elaborated on the effect of the tags on our results (line: 532-535, line: 542-545, line: 574-582, line: 586-596). Briefly, there was collinearity between tag type and tag weight, so we excluded tag type and included tag weights in our models. We found no statistical effect of tag weights on the dominant foraging strategy, the number of prey attacks or the success ratios. Furthermore, by relying on model selection procedures (in R using the “dredge” function), the best fitted models (using the AICcriterion) did not include any of the three tag explanatory variables. For the subset of data, where we retrieved the tags directly and, therefore, have recapture weights, we added change in body weight as an explanatory factor. There was no evidence that the loss in body weight was correlated with foraging strategies.

Figure 1DE As I commented in Table S1, TagB is heavier than Tag A, this may be the reason why gleaning is used more often in the foraging strategy.

Thanks for pointing this potential relationship out, we agree it is a relevant parameter to investigate. We have run different analyses to test for the different types and weights of tags in relation to the chosen foraging strategy. There is no significant relationship with the chosen foraging strategy (p-values>0.8) and the tag weights. Thus, it seems that tag sizes between 10-15 % of the body weights of the bats do not have differential effects on the chosen foraging strategy of the bats. This is an important control for the findings, so we have included this relationship in our statistical model and elaborated on the findings (line: 574-596)

Figure 2 As mentioned in the discussion, in addition to weather, what is the relationship between factors other than habitat in individuals that switch between gleaning and hawking relatively frequently and on foraging days? What about the relationship between a flight time from nest emergence and the number of food obtained up to that point (not only the actual number of successes but also the relationship with success rate)?

We greatly appreciate the ideas of how we could continue testing our data for new patterns to understand the foraging behavior of these animals, and we find your suggestions interesting to pursue. However, in the interest of revising the manuscript to make it simpler and clearer as requested by all reviewers, we have decided to keep the manuscript and the statistical model testing as it is.